# Ponte: Represent Totally Binary Neural Network Toward Efficiency

**DOI:** 10.3390/s24206726

**Published:** 2024-10-19

**Authors:** Jia Xu, Han Pu, Dong Wang

**Affiliations:** 1Institute of Information Science, Beijing Jiaotong University, Beijing 100044, China; timxu826@163.com (J.X.); ipuhan@bjtu.edu.cn (H.P.); 2Beijing Key Laboratory of Advanced Information Science and Network Technology, Beijing 100044, China; 3Intel Flex, Beijing 100091, China

**Keywords:** binary neural networks, computational efficiency, FPGA implementation

## Abstract

In the quest for computational efficiency, binary neural networks (BNNs) have emerged as a promising paradigm, offering significant reductions in memory footprint and computational latency. In traditional BNN implementation, the first and last layers are typically full-precision, which causes higher logic usage in field-programmable gate array (FPGA) implementation. To solve these issues, we introduce a novel approach named Ponte (Represent Totally Binary Neural Network Toward Efficiency) that extends the binarization process to the first and last layers of BNNs. We challenge the convention by proposing a fully binary layer replacement that mitigates the computational overhead without compromising accuracy. Our method leverages a unique encoding technique, Ponte::encoding, and a channel duplication strategy, Ponte::dispatch, and Ponte::sharing, to address the non-linearity and capacity constraints posed by binary layers. Surprisingly, all of them are back-propagation-supported, which allows our work to be implemented in the last layer through extensive experimentation on benchmark datasets, including CIFAR-10 and ImageNet. We demonstrate that Ponte not only preserves the integrity of input data but also enhances the representational capacity of BNNs. The proposed architecture achieves comparable, if not superior, performance metrics while significantly reducing the computational demands, thereby marking a step forward in the practical deployment of BNNs in resource-constrained environments.

## 1. Introduction

Binary neural networks (BNNs) have been increasingly recognized for their exceptional ability to alleviate storage constraints and expedite inference time. Due to the fact that the bit width of the binary neural network is only 1 bit, and its convolution operation is special as it only uses 1-bit XNOR and pop-count operation, it is often deployed on field-programmable gate array (FPGA) rather than general-purpose computing units to obtain better performance and reduce hardware logic area [1,2,3,4]. Recently, Compute-In-Memory (CIM) is a new computing paradigm that addresses the memory-wall problem in hardware accelerator design for deep learning. XNOR-RRAM [1] and 3T2RnvCIM [5] show good compatibility with BNNs with the emerging platforms.

However, there are still limitations on the BNNs. First, the first and last layers are typically full-precision, causing practical implementation challenges. Either they need to be calculated on a general-purpose computing unit or another dedicated computing unit for full-precision convolution, resulting in high computational resources or huge latency. For example, in [6], around 30% of the DSP slices are used by the first layer to perform fixed-point multiplication. In [7], it is necessary to repeatedly transmit data between FPGA and the CPU, which will cause a significant amount of delay and waste of bandwidth resources. Second, there is information loss in the concatenation between the first and second layers. Although the weight and input/output activation of mainstream BNNs are full-precision, the output of the first layer (typically 32-bit float-point [1,2,4,8] or 8-bit fix-point [9,10]) is truncated to a 1-bit value by the sign function, discarding most information. This information loss will be discussed from an information entropy perspective in Section 3.1. A BNN performs well on synthetic aperture radar (SAR) images, which show smaller gaps with real-valued neural networks [11]. Ref. [12] shows that the images are simpler and more likely to be binary images compared to RGB images, which supports that the truncation of the low-level features is still a critical problem in BNNs.

To address these issues, a binarized first layer is considered the most suitable implementation. Inspired by the fact that compressing the first layer to 8 bits fixed-point does not cause accuracy loss for the overall model [9,10], there are also some approaches [3,13,14] trying to avoid information loss by utilizing the 8-bit pixel input of the first layer while making the first layer binarized. It seems obvious to regard each 8-bit fixed-point output activation as 1-bit activations and regard each bit as an element of the input of the second layer (which is binary). However, our experimental findings indicate that the above approaches [3,13,14] encounter a significant drop in accuracy when applied to larger datasets such as ImageNet, suggesting that the modified layer architecture lacks sufficient capacity to represent or compress the input features, given that network models are essentially performing information compression  [15].

One potential explanation for the enhanced performance of full-precision convolution over its binary counterpart is that the multiplication operation in full-precision convolution introduces a higher level of non-linearity. In contrast, binary convolution is limited to the XNOR operation [1], which may not provide sufficient complexity for specific tasks. It is worth considering that the binary operation akin to a support vector machine (SVM) is relatively straightforward since the second layer’s input is restricted to binary values of +1 or −1. However, the issue arises when the weights are also binarized, effectively constraining the position of the SVM’s decision’s hyperplane. This constraint implies that the binary hyperplane lacks the necessary curvature to adequately separate all elements in the feature space into distinct hemispheres. At this juncture, the only recourse for an SVM is to employ dimensionality expansion [16]. By projecting the data into a higher-dimensional space, we can potentially restore the capacity for a more complex and effective separation of the classes, which is otherwise compromised in a binary setting.

To binarize the first and last layers of the BNN effectively, we propose a Represent Totally Binary Neural network Towards Efficiency (Ponte) method to design fully binary layers of BNNs carefully. Specifically, our proposed method involves a new coding technique called Ponte::encoding, which is dedicated to flattening the optimized landscape. A channel duplication method called Ponte::dispatch covers the ignored associated weight of each binary digit caused by the replacement of XNOR from 8-bit multiplication. Moreover, to further elevate the feature representation capacity of BNNs, we also proposed a method called Ponte::sharing, with duplicated channel numbers and a shared latent weight across the binary weights, thereby enhancing the representative capability of the binarized layer.

The primary contributions of this study can be encapsulated as follows:We examined the feasibility of employing binarized layers as the first and last layers in existing BNNs.We identified the challenges and limitations of binarized layers compared to full-precision layers, particularly in terms of non-linearity, convexity, and capacity.We proposed to binarize the first and last layers of existing BNNs using our contrusted optimization technique, Ponte. This approach maintains the same level of accuracy while reducing computational and storage workload.

## 2. Related Work

**Binary Neural Networks.** The foundational research on BNNs [9,17] laid the groundwork for an end-to-end training process for discrete networks. Courbariaux et al. [9] implemented a binarization of weights and activations through the application of the sign function, which resulted in negligible accuracy degradation on smaller datasets such as MNIST [18], SVHN [19], and CIFAR-10 [20]. However, the initial binarization of AlexNet [21] yielded a modest 36.1% Top-1 accuracy on the ImageNet [22] dataset. Consequently, subsequent research has predominantly been aimed at improving accuracy. The introduction of XNOR-Net  [1], which incorporates real-valued scaling factors with binary weight kernels, marked a significant advancement in binarization methods, elevating the Top-1 accuracy to 51.2% and reducing the performance disparity with the real-valued ResNet18 to approximately 18%. Building upon the XNOR-Net framework, Bi-Real Net [2] introduced the integration of shortcuts to facilitate the propagation of real values across feature maps, further elevating the Top-1 accuracy to 56.4%. MeliusNet [8] presents an architectural approach that consists of alternating a “Dense-Block”, which enhances the feature capacity, and an “Improvement-Block”, which improves the feature quality. Based on the observation that the performance of BNNs is sensitive to variations in activation distribution, ReActNet [4] proposes generalizing the traditional Sign and PReLU functions, denoted as RSign and RPReLU for the respective generalized functions, to enable explicit learning of the distribution reshape and shift at nearly zero additional cost. More recent efforts to enhance binary network performance have explored avenues such as broadening the channel width  [21], deepening the network architecture [9], and employing multiple binary weight bases  [23]. While these approaches have contributed to accuracy improvements, they have concurrently increased computational demands, diminishing the inherent compression benefits of BNNs. The ReBNN (resilient binary neural network) [24] is based on suppressing the frequent flipping of weights effectively. It attributes the frequent flipping of weights to a non-parametric weight scaling factor and then introduces a parameterized scaling factor and a weighted reconstruction loss for adaptive training. BiPer (binary neural networks using a periodic function) [25] replaces the traditional sign function for binarization by using a periodic binarization function. Specifically, a square wave is used to binarize weights and activations in forward propagation, and a sine function with the same period as the square wave is used as a differentiable substitute in backward propagation. Design (Designed Dithering Sign Activation) [26] applies multiple thresholds in the sign function Sign and follows the dithering principle to offset the sign function Sign for each pixel according to a spatially periodic threshold kernel. This method takes advantage of spatial correlation. Instead of quantizing each feature independently, it defines a joint offset for a group of adjacent pixels. A notable limitation of current BNNs in the context of hardware design is the retention of full precision in the first layer.

**Totally Binary Network.** To quantize the input layer of BNNs, Hirtzlin et al. [23] suggest employing stochastic computing to binarize the input images. This approach inflates the three input color channels from CIFAR-10 images to over 1500 binary channels, resulting in an approximately 16-fold increase in the number of parameters and Multiply–Accumulate operations (MACs) in the input layer. Dürichen et al. [9] explore two alternative strategies. The first involves utilizing an 8-bit fixed-point representation of a pixel, referred to as DBID. However, this method results in the loss of the associated weight of each binary digit upon conversion to a binary vector. The second strategy introduces a point-wise convolution layer between the images and the input layer while employing DBID. Essentially, this technique involves decomposing an 8-bit fixed-point number into eight 1-bit channels using binary-coded decimal (BCD) encoding. Nonetheless, this method does not resolve the inherent issues associated with DBID. Regrettably, when these two methods are implemented in the VGG-8 model on the CIFAR-10 dataset, they lead to a degradation in accuracy of at least 4.6%. FracBNN [14] enhances the architecture of ReActNet [4] by re-configuring network blocks and introducing a dual 1-bit activation scheme to bolster feature learning. FracBNN achieves competitive Top-1 accuracy on the ImageNet dataset, rivaling the full-precision MobileNetV2. BCNN [27] presents a tailored structure for ImageNet classification that boasts a smaller model size than MeliusNet and ReActNet. BiMLP [28] proposes a binary vision multi-layer perceptron architecture. On the other hand, FracBNN introduces a thermometer encoding that is sparsely encoded, which means that not all codes are utilized during conversion, leading to inefficiencies in encoding. The ILB (Input Layer Binarization) method  [29] presents a new method using 8-bit input data representation via bit-planes encoding and re-weighting to achieve the binarization of the first layer. The CIL (class-incremental learning) method  [30] extends prior work on incremental learning to BNNs by proposing a specifically designed fully-binarized network. HyBNN [31] shows that state-of-the-art (SOTA) BNNs like FracBNN and ReActNet often have to incorporate various auxiliary floating-point components and increase the model size to achieve satisfactory accuracy gains. However, this degrades hardware performance efficiency. HyBNN aims to quantify this hardware inefficiency in SOTA BNNs and mitigate it with negligible accuracy loss.

However, the DBID option is not differentiable between the binary convolution operations in BCD encoding and the 8-bit multiplication (MAC) operations. In DBID, each bit carries equal weight in binary operations, whereas, in 8-bit multiplication, higher-order bits contribute more significantly to the convolution sum, thus holding greater weight. Although this issue is addressed in FracBNN, the solution involves merely decoding BCD encoding, resulting in an excessive number of XNOR operations and an expanded channel size. Moreover, thermometer encoding merely decomposes values, artificially imposing a semantic wherein more “1” s in a channel signifies a more significant number. During training, each 255 channel corresponding to the RGB channels is associated with 255 independently optimized weights, oblivious to the thermometer encoding, complicating the training process. In our approach, we devised a novel encoding strategy that utilizes a limited number of channel replications and encoding to create a convex weight optimization space as much as possible, addressing the issue of blurred operational weights in DBID and reducing computational demands compared to FracBNN.

Furthermore, DBID and FracBNN only binarized the first layer, while the final layer remains full-precision. In our work, we apply the same binarization method to both the first and last layers, transforming the network into a fully binary one. This not only reduces the hardware resource overhead caused by the need for full-precision multipliers during inference  [32], but also thoroughly validates the effectiveness of our method.

## 3. Methodology

In this section, we will discuss the details of our contribution in the order above. Figure 1 illustrates the overall architecture of the proposed Ponte framework, which will binarize the first and last layers of BNNs. Our proposed architecture consists of three key parts while performing forward propagation before accessing the binarized first and last layers. The first operation is Ponte::encoding, which is a bit encoding method for both input activation (which is input pixel in 8-bit format) and the corresponding binary weight (grouped by eight and one-to-one correspondence with input activation). The role of bit splitting is to separate the encoded binary vector and decompose an n-dimensional binary vector into n channels value. The second operation is Ponte::dispatch, which is demonstrated as channel dispatch in Figure 1; other than conducting an operation on a pixel, which is a binary vector, the channel dispatch regards each bit as a single weight to perform the Ponte::dispatch operation. The third operation is Ponte::sharing, which demonstrates as latent-weight sharing in Figure 1. Specifically, we introduce the improvements of these three aspects in more detail.

### 3.1. Revisit on First Layer Binarization

In BNNs, it is common practice to leave the first convolution layer un-binarized. In layers beyond the first and the last layers, binary networks precisely provide a 1-bit input activation to the subsequent layer through a sequence of XNOR and pop-count operations, as shown in Equation (Equation 2). Here, in the first layer, all input pixels are processed using 8-bit weights to produce 8-bit feature maps, which are then truncated through the Sign function as illustrated in Equation (Equation 3) before being passed to the second layer. Here, for ease of reading, we denote the notation ∗ as below: (1)$$a*b=\sum_{i}a_{i}XNORb_{i}$$

Given a, b is a vector and $a_{i}$, $b_{i}$ is the i-th component of the respective vector.
(2)$$X_{b}*W_{b}=\mathrm{popcount}\left(\mathrm{XNOR}\left(X_{b},W_{b}\right)\right)$$
(3)$$x_{b}=\mathrm{Sign}\left(x_{r}\right)=\left\{\begin{array}{c} +1,\;\mathrm{if}\;x_{r}>0\\ -1,\;\mathrm{if}\;x_{r}\leq 0 \end{array}\right.,\;w_{b}=\frac{\left|\right|W_{r}{\left|\right|}_{l_{1}}}{n}\mathrm{Sign}\left(w_{r}\right)=\left\{\begin{array}{c} +\frac{\left|\right|W_{r}{\left|\right|}_{l_{1}}}{n},\;\mathrm{if}\;w_{r}>0\\ -\frac{\left|\right|W_{r}{\left|\right|}_{l_{1}}}{n},\;\mathrm{if}\;w_{r}\leq 0 \end{array}\right.$$
where $W_{b}$ and $X_{b}$ represent the matrices of binary weights and binary activations. $\left|\right|W_{r}{\left|\right|}_{l_{1}}$ is the $l_{1}$-normalization which makes the binary weight value $w_{b}$ close to the real weight (latent weight) value $w_{r}$.

The binarization process for both weights and activations is conducted using a Sign function. This implies two things: (1) There is a loss of information when transitioning from the first to the second layer. Therefore, without altering the second layer, the computational load of the first layer can be reduced to avoid generating excessive data. (2) The first layer differs from the others in employing real-value (8-bit or FP32) multipliers. In contrast, the binary XNOR and pop-count operations utilize entirely different hardware logic, which significantly impacts FPGA deployment. As illustrated in Figure 2, in traditional BNNs (with first and last layer un-binarized), the integration of additional computational units in FPGA for these un-binarized operations leads to resource wastage, which could be mitigated by adopting binary convolutions. Studies such as  [33] demonstrate that up to 30% of FPGA resources could be conserved, thereby enhancing inferential performance. Collectively, these findings underscore the potential benefits and practical relevance of fully binarized neural networks on FPGAs.

Therefore, it is ideal to design a network layer that can make the 1-bit activation its output, which will cause no data loss. Specifically, we can conclude that the compression ratio by applying the sign function is up to 16× from a 16-bit float point value to a 1-bit value. To calculate the information entropy of output activation of the 8-bit convolution, let us assume that the binary encoding is uniformly distributed across the possible range of float16 values. In this case, the entropy would be maximal since each value is equally likely. After applying the Sign function f(x) to the vector $l_{1}$, the resulting vector $l_{2}$ will only contain two possible values: +1 or −1. The entropy of this distribution can be calculated using the formula for binary entropy: (4)$$H\left(X\right)=-p\left(1\right)*log_{2}\left(p\left(1\right)\right)-p\left(-1\right)*log_{2}\left(p\left(-1\right)\right)$$
where p(1) is the probability of +1 and p(−1) is the probability of −1 in the vector l2. After applying the Sign function, we only need 1 bit per value (since there are only two possible outcomes). The compression ratio CR is given by
(5)$$\begin{array}{cc} CR & =\frac{H(X_{fp})}{H(X_{bin})}\\ \; & =\frac{16\times l1}{1\times l2}=\frac{16}{1}=16 \end{array}$$

Following the motivation above, it is easy to think that using the binary layer as the first layer is a suitable choice. For computer vision tasks, the input is often RGB images, which can be considered as the 8-bit fixed-point integer. It is easy to convert the input to an input type that the binarized layer can accept. As mentioned, DBID simply splits each pixel into a binary vector in order, which can not only convert the input to binarized but also cause zero data loss. Although it causes a significant amount of accuracy loss in large datasets due to its inherent defects, it is a good start for us.

### 3.2. Ponte::encoding

In the previous section, we discussed how, according to  [13], significant accuracy loss can occur on large datasets, one reason being that the output is not a monotonic function of either the input or weights. After decomposing an 8-bit input into a binary vector, which is eight 1-bit channels, the partial sums resulting from the operation of the 1-bit values with their corresponding weights do not monotonically increase with an increase in the original 8-bit input, i.e., for each input pixel p∈[0,255], p∈Z, the binary vector b of which represents
(6)$$Bin\left(p\right)=b=\left\{b_{0},\ldots ,b_{7}\right\},where\;p=\sum_{n=0}^{7}2^{n}\times b_{n}$$
where Bin(·) denotes the function that turns an input pixel into a binary vector, where b represents the binary vector; here, all b which is converted from the origin 8-bit value *p* forms a set named
(7)B={b|b=Bin(p),p∈[0,255]}

There, all the binary vectors b which map from every input pixel form the set B.

But given $\exists p_{1},p_{2},p_{3}$, we denote $p_{2}>p_{3}$; and $b_{1}=Bin\left(p_{1}\right),b_{2}=Bin\left(p_{2}\right),\;b_{3}=Bin\left(p_{3}\right)$, though $p_{1}\times p_{2}>p_{3}$, there is still some case which makes $b_{1}\;*\;b_{2}<b_{3}$. This behavior is inconsistent with that of real-valued networks and can significantly increase the difficulty of optimization during back-propagation.

To address this, we propose a coding method, Ponte::encoding, illustrated in Figure 1, specifically designed for mapping full-precision bits to binary values, which can generate a flatter function curve of the output with respect to the input and weights. The core of the method is to create a hash table, mapping each 8-bit input pixel to another 8-bit binary vector between the output of the predecessor layer and the input of the successor layer.

Specifically, since the input of the first layer is 8-bit, the encoding needs to rearrange the 256 combinations of 8-bit values to generate an affine relationship between the pre-encoded 8-bit values and the post-encoded eight 1-bit channel values. We follow the arrangement rules below, for which we define: the 8-bit precision input set A, the same definition of set B. Moreover, our goal is to find a function H(·) that can perform the best mapping from any b to the split channels set B, and $\bar{H}\left(\cdot \right)$ to mapping reversely from B to A. We define the symbol ≻ as the partial order relation among elements in the set. Let us set H:B→A as a function. For any a,b∈B, we define
(8)$$\begin{array}{c} a≻b,\;iff.\;H(a)>H(b). \end{array}$$
(9)$$\begin{array}{c} a≅b,\;iff.\;H(a)=H(b). \end{array}$$

Take an example; for the max value of an unsigned 8-bit integer, which is 255, the binary vector $b_{max}$ representation of which is {1,1,1,1,1,1,1,1}, and this value is most suitable for the max value in B because, for any $b_{t}$ in B and $b_{q}$ in B, we all have $b_{t}$∗$b_{max}≻b_{t}$∗$b_{q}$. This behavior is the same as the × operation in set A. This is the same as the min value 0 for both sets. But for a value 127 in set A, which corresponds to binary vector, and a value 128 to $b_{128}=\left\{1,0,0,0,0,0,0,0\right\}$, for any $b_{t}\in B$, $b_{t}$∗$b_{128}≻b_{t}$∗$b_{127}$, this conclusion does not hold. Therefore, we reiterate our purpose of finding the best ordering from A to B such that, for any $a_{t}\in A,a_{q}\in A$, there are as many b as possible that comply with our goal. We attempt to encode B as follows: (10)$$\begin{array}{cc} \; & if\;\sum_{i=0}^{7}b_{ti}>\sum_{i=0}^{7}b_{qi},\;then\;b_{t}≻b_{q}. \end{array}$$(11)$$\begin{array}{cc} \; & if\;\sum_{i=0}^{7}b_{ti}=\sum_{i=0}^{7}b_{qi},\;then\;if\sum_{i=0}^{7}b_{ti}\times 2^{i}>\sum_{i=0}^{7}b_{qi}\times 2^{i},then\;b_{t}≻b_{q}. \end{array}$$

Following these constraints, we sort all possible values of $B_{n}$ to obtain the mapping from A to B as $\bar{H}\left(A\right)=top\left(A,B,≻\right)$, where top(n,S,sym) is the *n*-th largest number in set *S* when using sym as the comparison method.

To illustrate the output convexity of our proposed Ponte::encoding, we carve out the optimization curvature of different methods. As illustrated in Figure 3, Figure 3a shows the optimized curvature of an 8-bit dot product between the input activation and weights. Figure 3b shows the optimized curvature of adopting the binary-coded decimal (BCD) [34] encoding method, which is a class of binary encoding of decimal numbers wherein each digit is represented by a fixed number of bits. Figure 3c shows the optimized curvature of the optimized curvature of fully adopting the Ponte method, which is smoother than that seen in Figure 3b. By comparing Figure 3b with Figure 3a, we can conclude that multiplication operations have a smoother function surface than binary convolution operations, and the function exhibits good monotonicity with the origin as the dividing plane. This provides a favorable optimization space for back-propagation. By comparing Figure 3b with Figure 3c, it can be concluded that Ponte::encoding can significantly compensate for the issues arising from replacing real-value multiplication with binary convolution in BNNs.

### 3.3. Ponte::dispatch

As Ponte::encoding makes the optimization curve smoother, if we dive into the bit-splitting problem, we will discover that each binary digit has its own associated weight in the 8-bit convolution. The weight information is lost when a pixel is converted into a binary vector after bit splitting. The magnitude of each bit becomes the same. As illustrated in Figure 4, if the associated weight is missing when computing, each bit’s change is equal for the binary operation but will cause a different value change stride in its original value representation. An additional insight gleaned from the DBID approach is the inherent difficulty in training a neural network to ascertain the weights corresponding to binary digits in order to facilitate an equivalent transformation. This aspect of BNN training presents a considerable challenge, underscoring the complexity of achieving effective weight learning within the constraints of binary representation.

To address the above challenge, as illustrated in the Ponte::dispatch diagram in Figure 1, we introduce a strategic replication of higher-significance binary channels. This means that binary channels representing more significant bits in the original 8-bit value are duplicated, effectively increasing their weight in the convolution process. Such replication is not uniform but is instead tailored to the significance of each bit, with more significant bits being replicated more frequently. For example, as illustrated in the Ponte::sharing diagram of Figure 1, the Least Significant Bit (LSB) is duplicated once, and the Most Significant Bit (MSB) is duplicated four times.

Our Ponte::dispatch algorithm is presented in Algorithm 1. We employ a channel duplication strategy, wherein when converting an 8-bit number into a binary vector, we replicate channels in ascending order from lower to higher significance. Channels corresponding to higher-significance binary digits will receive more duplicated channels. We explored various duplication strategies and selected one that balances computational load and precision. Specifically, the sequence is constructed such that the *i*-th term $S_{i}$ is the integer *n* for which the cumulative count of integers from 1 to *n* is at least *i*. The sequence is non-decreasing, and each term corresponds to a triangular number at its final occurrence. The n-th triangular number, which represents the position of the last occurrence of *n* in the sequence, is given by the following formula: $T_{n}=\frac{n(n+1)}{2}$. And to find the value of the i-th term in the sequence $S_{i}$, we seek the smallest integer *n* such that *i* is less than or equal to the n-th triangular number $T_{n}$. This can be determined by solving the quadratic inequality $\frac{(n-1)n}{2}<i\leq \frac{n(n+1)}{2}$. Thus, we have the following:

For given a pixel intensity p, i∈{1,…,L} is the index of its origin binary vector; thus, the dispatched vector TV index will be $i_{d}\in \left\{1,\ldots ,\frac{L*(L+1)}{2}\right\}$ where $TV\in {\left\{-1,1\right\}}^{\frac{L*(L+1)}{2}}$; then, TV is defined as
(12)$$n=\left\lceil \frac{-1+\sqrt{1+8i}}{2}\right\rceil$$

Namely, the number of duplicate channels is exactly equal to the index of the origin bit. The integer *L* is the dimensionality of the origin binary vector, which represents the exact pixel. The actual operation is described in Algorithm 1. An input image with RGB channels is now converted from 255 × 255 × 3 channels to 255 × 255 × 108 channels. Consequently, the dimensionality of the dot products increases, which substantially addresses the issue of associated weights on each bit.
**Algorithm 1:** The proposed Ponte::dispatch algorithm**Input**: original weight before expanding channel $W_{i}$, scale factor *S***Output**: weight with expanded channel $W_{o}$1:The expanded channel number of MSB$C_{m}=8*S$.2:The total channel number of expanded channel $C_{t}=\left(1+C_{m}\right)*8/2$.3:define offset_calc(i)=((1+i)×i)/24:for *i* in range(num_bits):5:   base offset of each bit’s expanded channel $O_{b}=offset\_cal\left(i\right)$6:   for *j* in range(i+1):7:      $W_{o}\left[:,O_{b}+j\right]=\left(W_{in}\left[i\right]>>i\right)\&1$ 8:   endfor9:endfor

### 3.4. Ponte::sharing

In this section, we will discuss in detail an innovative enhancement to the representational capability of BNNs named Ponte::sharing, which aims to address the issue mentioned in the Introduction Section regarding the inability of binary weights to provide complex classification representational capabilities. Specifically, BNNs produce outputs of only +1 and −1, leading to a scenario wherein all activations are considered to contribute equally to the computation, a situation that does not occur in real-valued networks. This equal contribution can prevent the network from effectively masking out unnecessary features, thereby interfering with further inference. By doubling each channel, we enable a single activation to correspond to multiple weights in the computation, thus generating more possible values (during the channel doubling process, zeros are introduced) and allowing for the masking of certain feature maps.

In our approach, we initially split the channels into two parts: (13)$$X_{b}=\left\{x_{\left\lfloor b/2\right\rfloor }|b\in \left[0,2N\right]\right\}$$
where *N* represents the number of channels. However, each weight of the added channel number will receive the same gradient when training, which will cause the value (and latent weights) to be the same, deprecating the representation ability. Therefore, we employ a training method that uses shared latent weights to achieve the desired effect of added channel numbers. The specific approach involves using the same latent weight to control two weights in the added channels, as illustrated in Ponte::sharing in Figure 1. During the forward propagation, the process is governed by the latent weight. Correspondingly, we also double the channel number of activation; thus, the corresponding weight number will also be doubled. At this point, we group the weights in pairs, and the method for updating the weights is as follows: (14)$$\left[w_{b},w_{b+1}\right]=\left\{\begin{array}{c} \left\{+1,+1\right\},\;\mathrm{if}\;\frac{\left|\right|w_{r}{\left|\right|}_{l1}}{n}Sign\left(w_{r}\right)>B\\ \left\{+1,-1\right\},\;\mathrm{if}\;-B<\frac{\left|\right|w_{r}{\left|\right|}_{l1}}{n}Sign\left(w_{r}\right)<B\\ \left\{-1,-1\right\},\;\mathrm{if}\;\frac{\left|\right|w_{r}{\left|\right|}_{l1}}{n}Sign\left(w_{r}\right)<-B \end{array}\right.\;s.t.b\%2==0$$
where B is the boundary of the bit shifting and works as a hyper-parameter when training.

The hypothesis driving our approach is that channel number doubling, when combined with shared weight awareness during training, can substantially elevate the feature representation capacity of BNNs. This is achieved without compromising the advantages of BNNs, such as their low memory requirements and high computational speed. The shared latent weight centralizes the accumulation and distribution of gradients across the doubled channel numbers, ensuring a unified and harmonious learning process across the augmented input space.

### 3.5. The Back-Propagation for the Last Layer

DBID, FracBNN, and PokeBNN have only substituted binary values for the full-precision first layer, and it is understood that the first layer does not need to consider the differentiability of its output with respect to the input. However, in our proposed framework, we implement the binarized last layer instead of the full-precision layer in traditional BNNs in classification tasks. Therefore, unlike previous work, we need to establish a back-propagation function for our binarized last layer. We define the formula as follows: (15)y=D(H(F(x)))

We denote D(x) as the channel weight awareness function, which is Ponte::sharing, and H(x) as the weighted channel expand function, which is Ponte::dispatch, F(x) represent Ponte::encoding, the mapping operation, which is our proposed encoding method. So, in the back-propagation stage, we have
(16)$$\frac{\partial y}{\partial x}=\frac{\partial y}{\partial D}\cdot \frac{\partial D}{\partial x}=\frac{\partial y}{\partial D}\cdot \frac{\partial D}{\partial H}\cdot \frac{\partial H}{\partial x}=\frac{\partial y}{\partial D}\cdot \frac{\partial D}{\partial H}\cdot \frac{\partial H}{\partial F}\cdot \frac{\partial F}{\partial x}$$

For ∂D/∂x, other than using Sign function to binarize the latent weight, we define $\phi \left(w\right)=\bar{w}$, where $\bar{w}$ denotes the assigned 2 bits $\left[w_{b},w_{b+1}\right]$ and *w* denote the shared latent full-precision weight. The derivative of the Sign function is zero in almost all regions, which ostensibly renders it incompatible with the back-propagation algorithm, as the precise gradient of the cost function with respect to the quantities before the discretization (pre-activations or weights) would be zero. Similar to the un-shared weights, according to each element of the“Ponte::shared” layer, we have
(17)$$y=\Phi_{b}\left(\vec{x},\vec{w}\right)=\left(x,x\right)\times {\left(w_{1},w_{2}\right)}^{T}=\left(x,x\right)\times \phi \left(w\right)$$

The gradient of the operation $\Phi_{b}\left(\cdot ,\cdot \right)$ is almost non-existent across its domain, necessitating the use of a “pseudo-gradient” to obtain a gradient signal on the latent weights *w* [9,35]. In the most basic scenario, this pseudo-gradient $\Phi^{\prime }$ is derived by substituting the binarization process with an identity function during the backward pass: (18)$$\begin{array}{cc} \; & \frac{\partial \Phi }{\partial x}\approx \frac{\partial \Phi_{b}}{\partial x=w*L} \end{array}$$(19)$$\begin{array}{cc} \; & \frac{\partial \Phi }{\partial w_{bin}}\approx \frac{\partial L}{\partial w} \end{array}$$

In this context, *L* signifies the task loss associated with the output. The utilization of a pseudo-gradient in conjunction with latent weights facilitates the application of a diverse array of established methodologies to BNNs. This includes a variety of optimization algorithms (such as SGD, Adam, etc.) and regularization techniques (like L2-regularization, weight decay) [36,37,38].

As for ∂H/∂x, our forward propagation can be viewed as
(20)$$f\left(\vec{x},\vec{w}\right)=\sum_{i=0}^{7}\sum_{j=0}^{i}x_{i}\;\mathrm{XNOR}\;w_{i},x_{i}\in \vec{x},w_{i}\in \vec{w}$$

In this implementation, the associated weights of higher-significant bits contribute a greater proportion to the accumulated result (convolution operation). Therefore, if there is a large gradient for a higher-significant bit, it indicates a desire for the corresponding pixel to decrease (or increase) more rapidly opposite the gradient direction. Consequently, we conclude that, after splitting an input pixel into a binary vector containing multiple 1-bit positions, the more significant bits will receive larger update gradients. Thus,
(21)$$\frac{\partial L}{\partial w_{i}}=\sum \frac{\partial L}{\partial w_{bin,i}*i}$$

For ∂F/∂x, during the back-propagation, the Ponte::encoding, which is the mapping function is ignored, i.e., ∂F/∂(x)=1. This is known as the straight-through estimator (STE) [9]. The derivative of the encoding operation is transparent to other operations when running the back-propagation.

Moreover, in real-valued networks, we typically apply a softmax function to the network outputs to obtain scores, which are then normalized and used in a percentage-based voting system to select the Top-1 and Top-5 predictions. For a binary layer, an input with *n* channels, after undergoing a pop-count operation, has an output range of [0,n]. Experiments indicate that, for the input channels of the last layer in existing neural network architectures, the output range of a binarized final layer when producing outputs is sufficient for producing the right prediction.

## 4. Results

To evaluate the efficacy of the proposed methodologies, we undertake experimental analysis on the CIFAR-10 and ImageNet datasets. Initially, we delineate the dataset characteristics and outline the training strategy. Subsequently, we present a comparative assessment of the proposed binary networks and algorithms, examining both their accuracy and computational expenditure. We then analyze the effects of Ponte::encoding, Ponte::dispatch, and Ponte::sharing in detail in the ablation study.

### 4.1. Experiment Settings

The experiment was carried out on both CIFAR-10 [20] and ILSVRC2012 ImageNet [39] classification datasets. The CIFAR-10 datasets were used to demonstrate and compare the related works. The ILSVRC2012 is a more challenging job for a BNN model. In our experiments, the data augmentation method is described in [29] and widely used to control the variable.

Our approach adheres to the established binarization technique, as detailed in [4]. The procedure is executed in two distinct phases. Initially, we train a neural network featuring binary activations and real-valued weights from an uninitialized state. Subsequently, we utilize the weights derived from the initial phase as the starting point and proceed to fine-tune the network, this time with both weights and activations in binary form. Throughout both phases, we deploy the Adam optimization algorithm, complemented by a linear learning rate decay scheduling strategy. The initial learning rate is configured at 5 $\times \;10^{-4}$. The training extends over 600 k iterations with a batch size of 256. For the first phase, we apply a weight decay of 1 $\times \;10^{-5}$, which is then reduced to 0 during the second phase.

### 4.2. Energy Efficiency and Parameter Size Analysis

Ponte achieves a seven-fold reduction in computational energy consumption compared to an 8-bit first layer with a similar number of parameters and much less computational energy, while also delivering better accuracy.It is not meaningful to directly compare binary MACs with 8-bit MACs, as binary MACs consume significantly less energy in practical deployments [40]. PokeBNN has proposed a method for assessing energy efficiency.
(22)$$ACE=\sum_{0\leq i<I}a_{i}2^{i}\sum_{0\leq j<J}b_{j}2^{j}=\sum_{\begin{array}{c} 0\leq i<I\\ 0\leq j<J \end{array}}\left(a_{i}\wedge b_{j}\right)2^{i+j}$$

Based on this, we can deduce that the energy consumption of a binary MAC (measured in ACE [3] units) is 1/64th of that consumed by an 8-bit MAC. The Arithmetic Computation Effort (ACE) serves as a quantifiable cost metric for BNNs, enabling joint optimization of cost and accuracy. A variant of BNN demonstrated a threefold reduction in ACE while maintaining accuracy, highlighting the efficacy of ACE in guiding network optimization. This approach is beneficial in resource-constrained deployments wherein efficiency and accuracy are crucial.

We conducted calculations on three benchmark strategies on the ImageNet dataset using ResNet18 and compared the computational load of the first and last layers and the ACE values, as well as the parameter size, as shown in Table 1. From the calculations, we can conclude the following.

From the energy-efficient perspective, the quantization to 1-bit of layer plays a significant role on both memory consumption and computational demand, and the comparison result of memory consumption and computational demand between our proposed method with BirealNet [2] as a reference design and the previous work FracBNN [14] is illustrated in Table 1. For the first layer, we can observe from Table 1 that, although FracBNN replaces FP(float point) MACs with BMACs, eliminating the need to deploy the first layer on specialized proprietary hardware, the excessive number of channels and lack of encoding behavior lead to a substantial increase in computational load. This results in an energy consumption almost three times that of the traditional BirealNet convolution. This not only allows us to reuse hardware specifically designed for binary convolution without the need for 8-bit MAC hardware but also achieves a 5.3-fold reduction in computational energy consumption. This significantly reduces the hardware resources and energy consumption of the computing devices. For the last layer, we calculate the computational, memory consumption, and ACE based on the BirealNet [2] and our binarized layer, which replaces the last layer of [2]. For convenience, we take CIFAR-10 as an example, with 10 elements out (10 output channels) and 1024 channels input. Table 1 provides details of this. Both BirealNet and FracBNN use the float point layer as their last linear layer. This not only allows us to realize a fully binary network but also achieves a 1.7-fold reduction in computational energy consumption compared with BirealNet.

From the Parameter size perspective, binary weights only require 1 bit to be stored, so each weight can save up to 32× the space compared to full precision. Due to the fact that BirealNet uses full-precision floating-point numbers consistent with [1] as the weights of the first layer, although it uses the least number of input channels, each full-precision datum still occupies 32 bits of storage space, so the memory space occupied is still 2.66× greater than that of our proposed method. In FracBNN, due to the higher channel number, the parameter count of the first layer is up to 20.5× higher, which also puts pressure on storage and bandwidth. The result is 10.5 times greater storage space than our method. However, with our encoding method, using the least number of parameters, which is only 2.5 MBytes, we replace all int8 MAC operations with BMACs, achieving the highest compression of storage. The last layer is also the same. We still use The BMAC to replace the FP MAC in the fully connected (FC) layer. Thus, FracBNN [14] does adopt the same strategy as BirealNet. They use the same full-precision FC as the last layer. By replacing the last layer with a binarized layer, we achieve 2 times compression from the reference design (BirealNet) by shrinking the parameter size from 40 MBytes to 20 MBytes and compressing the computational energy for 1.7 times with the measurement of ACE.

### 4.3. Comparison with State of the Art

We conducted experiments using BirealNet [2] as a baseline, and compared the performance of Ponte with the state-of-the-art works, the results of which indicate that Ponte achieves competitive performance on both CIFAR-10 and ImageNet, comparable to the benchmark. We also compared our method with the recently leading first-layer quantization approach, FracBNN, and achieved competitive inference results with significantly reduced computational energy consumption.

Firstly, we established our experiments on a model based on BirealNet [2], replacing the first and last layers of BirealNet with our proposed method. We compared the results on two datasets, CIFAR-10 [20] and ImageNet [39], under ResNet18 and ResNet34 architectures. For the ImageNet dataset, two sets of experiments, each employing distinct binary quantization schemes, as proposed in recent studies, were juxtaposed with the Ponte method in the accompanying Table 2. In particular, the first set of reference models includes DBID [13], XNOR-Net++ [41], and BirealNet [2], and all of the experiments were conducted using the BiReal quantization scheme. In the second set of experiments, we included ReActNet [4] and FracBNN [14], and both of the experiments used the ReAct quantization scheme. To show the absolute performance of our design, the performance of the real-valued model MobileNetV2 [42] was also introduced. Table 2 presents a comprehensive breakdown of the computational complexity associated with BMAC (Binary MAC) operations and FPMAC (Floating-point MAC) operations, the costs related to storage, and the recorded Top-1 and Top-5 accuracy scores.

In the first group of models, all of which implemented the BiReal quantization scheme, BiRealNet solely executed binary quantization on the model without incorporating additional optimizations to mitigate the reduction in the neural network’s capacity. XNOR-Net++ [41] improved accuracy by training BNNs by learning a single scaling factor for both weights and activations via back-propagation, which enhanced accuracy by over 6% on ImageNet datasets on ResNet18. The result from XNOR-Net++ illustrates the utilization potential of the binary neural network by revising the distribution of the activations. DBID is an intuitive method to directly convert the image data into a binary vector as the neural network without any data loss. We adopt the DBID [13] algorithm into the BirealNet scheme and obtain the baseline performance of the intuition algorithm with only 38.1% and 41.7% for ResNet18 and ResNet34, respectively. The severe drop in accuracy in the results shows that the binary neural network cannot fully utilize the data format directly converted from 8-bit data representation. A conclusion can therefore be drawn that finding the right way to represent the input data in a way that can be fully utilized by the binary neural network is necessary. We demonstrate the potential of our proposed method to optimize the data representation and illustrate its effectiveness by comparing it with the vanilla BiRealNet, which is implemented in the same experiment settings except for replacing the input layer into Ponte. The result shows that our proposed method increased the accuracy by 0.7% on Top-1 and 0.5% on Top-5 score on the ImageNet dataset in ResNet18 model architecture, with 75% computational efficiency improvement, all in BMAC. Similarly to the ResNet34 model, a huge step of computational efficiency improvement and a comparable Top-1 and Top-5 score are presented.

In the second group, which adopted the ReAct quantization scheme, ReActNet is a higher-performance baseline for us to further prove the effectiveness of our proposed method and to prove the agnostic model scheme, which is available to extend and deploy in different model designs and architecture. Our method maintains the same network architecture as the original ReActNet, ensuring a fair comparison in terms of structural complexity. Notably, our ReActNet variant demonstrates a significant performance improvement over the original ReActNet-A, achieving a Top-1 accuracy of 71.5% and a Top-5 accuracy of 88.9% on the benchmark dataset. This enhancement is achieved without any floating-point multiply–accumulate operations (FP MAC), which underscores the efficiency of our approach. Compared with FracBNN, which exhibits higher accuracy than our method, it owes its performance gains to architectural modifications that have been proven to yield a 3% accuracy improvement on CIFAR-10 with ResNet-20 on the ablation study. However, it is important to note that these modifications are not present in our ReActNet variant, which strictly adheres to the original ReActNet structure. This strategic substitution has resulted in a notable increase in accuracy without the need for additional computational complexity, as evidenced by the removal of FP MAC operations.

In addition to the advantages in target recognition accuracy, the computationally intensive full-precision matrix multiplication operations involved in the first and last layers can be replaced by lightweight bit-wise XNOR and bit-count operations. The removal of full-precision computations allows for the complete elimination of full-precision computing units during FPGA deployment, thereby optimizing the occupancy of logic resources. Furthermore, there is no need to balance the load between full-precision data computation and binary computation units, which will benefit the inference process and host accelerator interaction. Moreover, all computational processes of Ponte can be deployed without the addition of any dedicated computing logic. Therefore, we have reason to believe that deploying Ponte on devices such as FPGAs can achieve additional execution speed without compromising accuracy.

### 4.4. Ablation Study

To rigorously evaluate the impact of our proposed techniques on the performance of BNNs, we conducted a series of ablation studies. These studies were designed to isolate and understand the individual contributions of Ponte::encoding, Ponte::dispatch, and Ponte::sharing. Each technique was incrementally removed or modified in our BNN architecture to assess its effect on model accuracy and computational efficiency. The experiments were conducted on the CIFAR-10 and ImageNet datasets, a standard computer vision benchmark, to ensure our findings’ relevance and applicability.

#### 4.4.1. The Effectiveness of Ponte::encoding

As mentioned earlier, due to the unique nature of the XNOR operation, the computations after encoding do not satisfy the original partial order relationship, meaning that the result plane of the operation is non-convex. We have already verified this phenomenon both theoretically and with small examples. Now, we proceed to validate the performance of the encoding in a real neural network training scenario by conducting experiments on the CIFAR-10 dataset.

We conducted a comparison using the original 8-bit BCD encoding and the encoding method we proposed, as shown in Table 3, and found that our encoding significantly outperformed the traditional BCD approach in terms of maintaining the monotonicity of the activation function. This improvement was particularly evident when examining the behavior of the encoded values during the forward and backward propagation, wherein our encoding demonstrated a more consistent gradient flow.

Additionally, in order to prove the effectiveness of the purposed encoding, we conduct another experiment by using the existing order of a binary numeral system to represent the input pixel instead of Ponte::encoding. We chose the Gray code (also known as reflected binary) [43], Aiken code (also known as 2421 code) [44], and Stibitz code (also known as Excess-3, shifted binary, rem-3) [45] for comparison. The Gray code is a non-weighted code that is designed to prevent errors in digital communication by ensuring that only one bit changes at a time between successive values. The Aiken code is another weighted binary code system used for decimal digit representation, which is unique in its self-complementing feature, allowing for straightforward negative number representation. Lastly, the Stibitz code is a biased representation method that adds an excess number to the decimal value before converting it to binary; here, we set excess to zero to represent all non-negative numbers. For these three types of code, transform each pixel of the RGB image from BCD encoding to the code, respectively, and bypass the Ponte::encoding method to the Bit splitting phrase.

We compared our encoding scheme with the methods introduced above. As shown in Table 3, the results indicate that, while these traditional encodings may or may not offer certain benefits, Gray code and Stibitz code significantly reduce accuracy. This indicates that our proposed method does provide a superior balance between computational efficiency and representational fidelity by providing a smoother optimizing landscape.

Therefore, we conclude that the results visualized in our toy example and those obtained from experiments in real-world scenarios are highly consistent in their conclusions. Moreover, due to the unique properties of the XNOR operation, employing specific encoding to make the optimization plane smoother can achieve excellent results in training, making a remarkable impact on the accuracy of BNNs.

#### 4.4.2. The Effectiveness of Ponte::dispatch

Our second study examined the role of Ponte::dispatch, which is implemented as channel duplication, a method devised to amplify the influence of more significant bits in the input data. By varying the duplication factor across different bit positions and comparing these models to a baseline without channel duplication, we found that models employing strategic channel duplication exhibited improved accuracy. This improvement highlights the technique’s effectiveness in enhancing the model’s capacity to capture and represent critical information, thereby mitigating the limitations of binary convolution.

Furthermore, we conducted various experiments to verify the effects of 1. varying the number of channel duplication and 2. employing different channel duplication strategies, i.e., assigning different channel numbers across all the extended channels. The results are shown in Table 4. First, we explored different extended channel numbers to find the best channel numbers that can obtain the balance between accuracy and computational cost. From Table 4, the experiment shows that the accuracy is the highest at 56 channels with a Top-1 accuracy of 81.38%, and we can see that we may not gain more accuracy benefit with a higher number of channels. Secondly, an experiment with different copy numbers for different bits in a binary vector according to the given dispatch algorithms in a limited extended channel number is conducted. The linear is the aforementioned using triangular numbers to duplicate the channel for different bits. The exponential means that the copied channel number will increase exponentially according to the bit in the binary vector from low to high. As for the log method, the exponential means that the copied channel number will increase in log according to the bit in the binary vector from low to high. The result shows that the linear method achieved the best accuracy in the listed experiment. All experiments were carried out on the CIFAR-10 dataset.

#### 4.4.3. The Effectiveness of Ponte::sharing

The third ablation study targeted the weight gradient sharing technique, which allows for shared learning signals among duplicated channels, illustrated in Table 5. By disabling this feature, we aimed to assess its impact on the learning dynamics and overall model performance. The results indicate a degradation in model convergence and accuracy, emphasizing the significance of weight gradient sharing in promoting coherent learning across duplicated channels and enhancing the representational capability of the binarized layer.

To validate the consistency of weight sharing in Ponte::Sharing, we conducted two sets of experiments: one wherein channels are duplicated, but each binary weight has a separate latent weight, and another wherein duplicated channels share a single latent weight. The results are presented in Table 5. We observed that training without weight sharing leads to a significant decrease in neural network accuracy. It may not even yield a significant difference compared to not duplicating channel numbers at all.

Due to the fact that the boundary is a hyper-parameter, it is necessary to select a proper value for the boundary carefully. Firstly, we will establish a pre-selected boundary and then compare the convergence curves of the training loss. The clipping boundary B serves as a hyper-parameter and plays a crucial role in low-bandwidth quantization, yet it has seldom been explored in past BNN research. In most cases, B is defaulted to 1 by convention. However, since we have redefined the boundary to encompass three segments instead of the traditional two segments in BNNs, it is necessary to experiment with boundary B.

Our experimental result indicates that parameter B works considerably and influences the accuracy of BNNs. We experimented with different B values, spanning from 0.001 to 10 in multiplicative increments, and the outcomes are demonstrated in Table 5. Our hypothesis posits that an elevated clipping threshold enhances the Lipschitz constant of the loss surface and concurrently diminishes the prevalence of dead neurons, which are characterized by their zero gradients. This aligns with the conclusion from PokeBNN, which underscored the significance of multiple definitions of the clipping boundary in the context of ultra-low bit-width quantization.

We also conduct experiments on the loss curve of different boundaries. Selecting the value of the various boundaries from 0.001 to 10. The Top-1 accuracy on ImageNet is listed in Table 6, and Figure 5 presents the loss curves under varying boundary conditions in a visualized way. Upon analysis, it is evident that the curves corresponding to boundary values of 0.001 and 0.1 closely track each other throughout the training process, suggesting a similar learning behavior under these conditions. The blue curve, representing a boundary value of 0.1, starts with a slightly higher loss but converges to a similar value as the previous two after a sufficient number of iterations. This indicates that, while the initial sensitivity to the boundary condition may affect the early stages of training, the model can still achieve comparable performance with adequate training. In contrast, the green curve, with a boundary value of 10, maintains a consistently higher loss throughout the training process. This suggests that such a high boundary value may be sub-optimal for the model, potentially due to over-regularization or an inability of the model to adequately capture the underlying data distribution. In conclusion, the loss curves suggest that boundary values in the range of 0.001–1 are more conducive to model convergence and result in a lower loss, with 0.001 and 0.1 providing the most similar and optimal performance. The boundary value of 10 appears to be detrimental to the learning process, highlighting the importance of careful boundary selection in the context of model training dynamics.

## 5. Conclusions

In summary, this study explored the use of binarization of the first and last layers, which are traditionally full-precision within BNNs. This was motivated by the need to reduce the computational and storage requirements associated, and also reduce the complexity of designing the hardware with full-precision layers. We started with the analysis of information loss due to binary truncation and the non-convexity of the optimization sphere in the previous work and then proposed a new approach, trying to overcome these obstacles without compromising the model’s accuracy or increasing the computational load. As a result, we compared our proposed design with our baseline and the result shows that our approach achieves up to 5.3× compression of computation energy and up to 2.66× compression of parameter size on a single layer, which makes the BNN more energy-efficient and storage-efficient.

There are still some deeper topics that can be investigated. Firstly, since our intuition is partly inspired by the multiplication mechanism of the float-point number, it is necessary to construct a model that can show the maximum representation capability of a specific size of the binarized layer to guide the design of the size of our neural network. Secondly, a deeper network may gain the representation ability of a neural network [46], but may be harder to optimize [47], so we will also investigate the better configuration of layer width and depth design of the first and last layer in BNNs. We are still working on the high-energy efficient implementation. Firstly, we seek to explore the generalizability of the proposed approach. We plan to implement the method on different model tasks such as object detection, generative models, etc. Secondly, we plan to co-optimize with FPGA BNN architecture circuit design and graph-schedule optimization. Thus, we have mentioned that the proposed approach does not need the extra FPGA area dedicated to the full-precision convolution/matrix multiplication. The instruction scheduling and operation dependency can be rescheduled to gain a higher utilization and inference performance. Lastly, due to the discontinuity characteristic of the binarized layer, it brings many challenges to the convergence in training procedure and accuracy in inference, and it is especially serious in all binary networks. So, we think that the distribution statistics and the training technique dedicated to binarized networks are worth studying in the future.

## Figures and Tables

**Figure 1 sensors-24-06726-f001:**
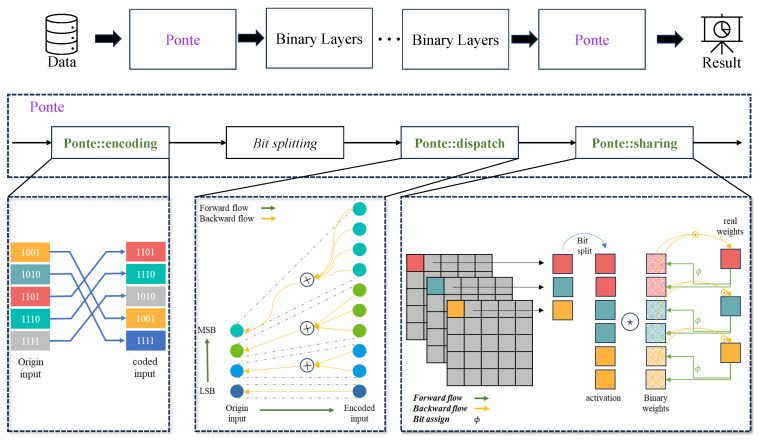
Illustration of the network architecture of Ponte, which consists of three key components: the Ponte::encoding, Ponte::dispatch, and Ponte::sharing. These are labeled in the dotted line box named Ponte and using green color in the middle of the diagram. The details of the components mentioned above are shown at the bottom of the diagram with respect to the aforementioned position from left to right.

**Figure 2 sensors-24-06726-f002:**
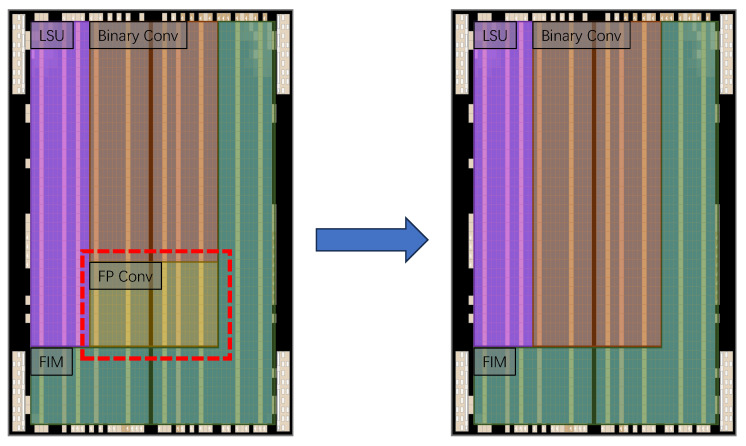
A concept schematic diagram of Chip plan in FPGA. The schematic diagram illustrates the chip plan for an FPGA with FP convolution-enabled BNN accelerator. The FP Conv logic occupies a large area with low active running states during a single inference.

**Figure 3 sensors-24-06726-f003:**
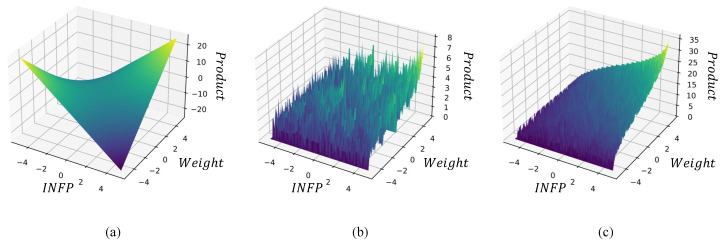
Output convexity compare. The INFP (input feature map) and Weights refer to the value of the 8-bit input feature map (images) and the value of an 8-bit vector, respectively, while the Product represents the binary MAC (BMAC) result of the INFP and Weight. (**a**) Optimized curvature of an 8-bit dot product. (**b**) Optimized curvature of adopting the BCD [34] encoding method. (**c**) Optimized curvature of adopting Ponte::encoding.

**Figure 4 sensors-24-06726-f004:**
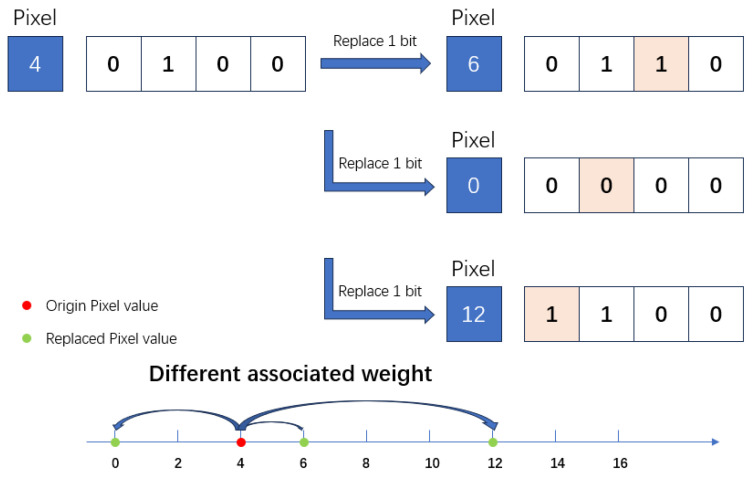
Problem without associated weights encoding. The figure shows the DBID-like algorithm mapping the fixed-point weights to the binary weight, which will cause different strides of value change by changing only one bit of the binary vector.

**Figure 5 sensors-24-06726-f005:**
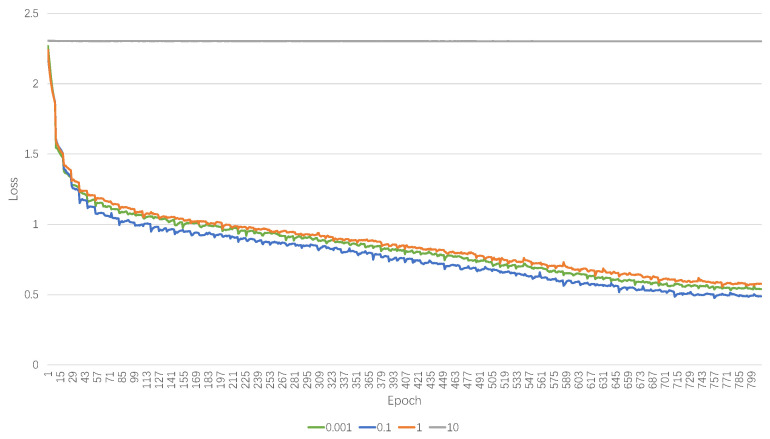
Loss curve on different boundary selections.

**Table 1 sensors-24-06726-t001:** Energy consumption and parameter size of first and last layer.

Layer	Method	Input Channel	Output Channel	Params (MBytes)	FP MAC	BMAC	ACE
First Layer	BirealNet [2]	3	64	6.75 (ref)	1.11 $\times \;10^{8}$	-	1.11 $\times \;10^{8}$ (ref)
	FracBNN [14]	256	64	36.0 (0.18×)	-	1.90 $\times 10^{10}$	2.97 $\times \;10^{8}$ (0.37×)
	Ours	72	64	2.53 (2.66×)	-	1.34 $\times \;10^{9}$	2.09 $\times \;10^{7}$ (5.31×)
Last Layer	BirealNet [2]	1024	10	40 (ref)	327,680	-	1.02 $\times \;10^{4}$ (ref)
	FracBNN [14]	1024	10	40 (1.0×)	327,680		1.02 $\times \;10^{4}$ (1.0×)
	Ours	36,863	10	20 (2.0×)	-	368,630	5.76 $\times \;10^{3}$ ( 1.7×)

**Table 2 sensors-24-06726-t002:** Comparison of the Top-1 and Top-5 accuracy with state-of-the-art methods on ImageNet dataset.

	Network	Model Size	$\mathrm{BMAC}\;(\times 10^{9})$	$\mathrm{FPMAC}\;(\times 10^{8})$	Top-1 (%)	Top-5 (%)
BDID [13]	ResNet18	2.53 (MB)	2.37	0	38.1	52.8
	ResNet34	3.79 (MB)	4.22	0	41.7	55.3
XNOR-Net++ [41]	ResNet18	5.10 (MB)	5.47	1.29	57.1	79.9
BirealNet [2]	ResNet18	4.2 (MB)	1.68	1.35	56.4	79.5
	ResNet34	5.46 (MB)	3.53	1.39	62.2	83.9
**Ours(Bireal)**	ResNet18	4.99 (MB)	3.02	0	57.1	80.0
	ResNet34	6.25 (MB)	4.87	0	62.5	83.8
ReActNet-A [4]	ReActNet	4.56 (MB)	4.83	0.12	69.4	-
FracBNN [14]	-	4.56 (MB)	4.56	0.01	71.8	90.1
MobileNetV2 [42]	MobileNetV2	3.47 (MB)	0	3.00	71.8	91.0
**Ours(React)**	ReActNet	5.82 (MB)	6.17	0	71.5	88.9

**Table 3 sensors-24-06726-t003:** Accuracy on CIFAR-10 for different coding methods.

Network/(%)	BCD	Gray	Aiken	Stibitz	Ours
ResNet18	80.05	47.45	74.11	52.40	81.38
ResNet34	85.38	49.84	75.02	55.79	88.22

**Table 4 sensors-24-06726-t004:** Accuracy on CIFAR-10 using different channel number assignment methods.

Method	8 (No Opt.)	28	56	84
linear	71.94	74.16	81.38	80.02
exponential	69.15	73.25	79.96	81.23
log	70.94	70.94	80.94	80.81

**Table 5 sensors-24-06726-t005:** Ablation study on weight sharing.

	w/o Sharing	Sharing
Top-1 (%)	77.62	81.38
Top-5 (%)	95.44	98.95

**Table 6 sensors-24-06726-t006:** Ablation study on different boundary selections.

B	0.001	0.01	0.1	1	10
Top-1 (%)	78.51	79.45	81.38	69.52	9.95

## Data Availability

Data are comtained within this article.

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
