# Peer review of "Ponte: Represent Totally Binary Neural Network Toward Efficiency"

_sensors, 2024, doi:10.3390/s24206726_

Round 1
Reviewer 1 Report
Comments and Suggestions for Authors
The manuscript contains the try to optimize the binary neural network for speeder and minor memory consumption. This subject is not entirely but indirectly could placed in the journal's scope.
Shortcomings:
1. Language should be correct:
- The authors use US and British spelling in the manuscript. You should not use both in one document.
- Capital letters are missing in many places.
- Problems with the subject in the sentences.
2. All abbreviations should be explained at first occurrence.
3. Some mathematical formulas used are not numbered.
4. Reference of tables in the text should be before the place of the table.
5. Table 7 is not referenced in the text. It means that this table is not necessary.
6. The first reference to figures in the text should be before the figure's place.
7. In the conclusion section, the authors should place their plans for the future in the manuscript's subject.
8. There are ancient items on the reference list. Only three items are from the past three years. This means that the research background was based on ancient manuscripts and that the novelty of the presented methods is doubtful and hard to assess.
9. Descriptions of the axis are blank on the Figure 7 charts.
10. I suggest changing the title of the article to more undesirable. This one is understandable only for industry-specific groups.
Comments on the Quality of English LanguageLanguage should be correct. Main problems are:
- The authors use US and British spelling in the manuscript. You should not use both in one document.
- Capital letters are missing in many places.
- Problems with the subject in the sentences.
Reviewer 2 Report
Comments and Suggestions for Authors
The paper proposes a new approach termed Ponte that allows extending the binarization process to the first and last layers in a binary neural network. The contribution is solid and the results achievs significant improvement.
Major:
As the architecture of layer Ponte is one of the main contribution of the paper, the detailed description of the architecture should not be presented in the introduction. Only th essence of the contrubution should be described in the introduction, but the details must be presented and justified in a separate section.
The layout presented in Fig 2 must be explained and any improvement presented and discussed.
In Section 4.2, the title is about memory usage analysis, yest in the text you present energy consumption. This is confusing!
In terms of what the energy consumption is computed? What is the metrical unit? This is not clear from performance results presented in Table 1 and Table 2.
The same can be said about the memory footprint. In terms of what the memory usage is evaluated? What is the unit? This is not clear from the presented results.
The authors state that proposed methodology achieves a 7-fold reduction in computational energy consumption compared to an 8-bit first layer with a similar number of parameters, and much less computational. However, this is not clear form the results presented in Table 1 and Table 2. The improvement factor must be defined and presented in these table taking one of designs as a a reference.
What are the limitation of the proposed methodology? Surely it is not perfect. How can it be improved. THis should be sstated and discussd in the second paret of the conclusion section, which should be about future work.
There is only one referenced work from 2023 and the rest of references is a decade old. The references and especially those about the related works must be updated by reviewing/citing recent related works form the last 2 to 3 years.
Minors:
The writing is somehow verbose. The authors should revise their text to avoid subjective qulification and restrict themselves to tecnichal description only.
The artwork of a figure must appear after the figure citation.
You should separate the "(" and "[" from the word immediately before.
Don't use bold face in the text: dataset; training strategy, Algorithm ..., weight sharing, etc.
Comments on the Quality of English Language
The writing is somehow verbose. The authors should revise their text to avoid subjective qulification and restrict themselves to tecnichal description only.
Reviewer 3 Report
Comments and Suggestions for Authors
This paper, “Ponte: rePresent Output of Neural network Towards Efficiency”.
My comments on this paper, such as the following:
The authors of the paper clearly explain the mathematical arguments that support the proposed method. The experiments that were carried out to demonstrate the efficiency of the proposed method were adequate. However, it would have been convenient to use more sets of images.
Reviewer 4 Report
Comments and Suggestions for Authors
Dear authros,
These are my comments regarding this manuscript.
There is no reference shown to support the first paragraph of the Introduction related to BNN.
A series of references are shown [1]], [2], [3] [4], it is suggested to describe them separately to see how each of them contributes to the manuscript, formatting errors are detected [[5], [6]].
It is mentioned that “There are also some approaches trying to avoid information loss of utilizing the 8-bit pixel, but it is not mentioned which ones they are.
In the stages, it is not described how Bit Splitting contributes to the Ponte architecture.
It is not suggested that figures be placed in the Introduction.
It is not demonstrated that the method to binarize images maintains a better precision and reduces the computational and storage workload, because these metrics are not presented or demonstrated.
It is recommended to organize the references in order of appearance and verify that they are all cited.
There is no complete study of the state of the art of related works in the three areas to improve precision and reduce the computational and storage workload.
Figure 3 does not describe what X1, X2, and Y represent.
Page 7 is incomprehensible and difficult to read. It is cluttered with constants and variables and out-of-format equations.
Algorithm 1 is incomplete, with no inputs or outputs.
Figure 5 has no units on the axes, and does not immediately state what it adds to the manuscript.
A better effort should be made to organize the paper and present the systems of equations.
Also, adequately describe the tests performed to affirm that the accuracy and reduction of computational and storage workload of the presented technique is superior to that of the state of the art.
Best Regards
Reviewer 5 Report
Comments and Suggestions for Authors
The presented research appears to be solid. However, there are certain request to improve the readability of the manuscript.
1. In the first three sections, there is often repetition of the text in terms of meaning. Please restructure the content to better group information about the proposed approach.
2. The mention of Gray code and other established methods is only on page 16 of the manuscript, moreover, without references to the works. A conceptual comparison is desirable in section 3.
3. Please indicate what specific FPGA hardware was used in your research.
4. Please provide an analogue of Figure 2 for BNN using the the proposed approach.
Round 2
Reviewer 1 Report
Comments and Suggestions for Authors
The authors corrected almost all shortcomings but didn't correct the language well. I suggest using a pro spelling checker or native speaker help.
Background research is good and acceptable, but it could still be better.
Asessment of the quality of the presentation is lower due to language problems.
Comments on the Quality of English LanguageI suggest using a pro spelling checker or native speaker help.
Author Response
Dear Reviewer,
Thank you very much for taking the time to review this manuscript. Please find the detailed
responses below and the corresponding revisions/corrections highlighted/in track changes in
the re-submitted files.
Q1: The authors corrected almost all shortcomings but didn't correct the language well. I suggest using a pro spelling checker or native speaker help.
R1: Thanks a lot for your advice. We have checked our manuscript thoroughly and corrected the spelling that is wrong, and rephrased the sentences as possible. Some critical rephrased sentences and words are highlighted in red in the revised manuscript.
Q2: Background research is good and acceptable, but it could still be better. 
R2: Thank you for your constructive feedback regarding the background research presented in our manuscript. We have taken your comments into consideration and have expanded our background in application prospects to the emerging chips for BNNs, the issue for implementation in previous work, and identifying the gap. We have also included additional recent studies and added supplementary illustrations in the introduction part, highlighted in red in line 24-27, 32-34 in page 1, and line 39-43 in page 2.
Best Regards,
Jia
Reviewer 2 Report
Comments and Suggestions for Authors
The authors has treated all my concerns reasonably. I suggest accepting the paper as is.
Author Response
Dear Reviewer,
Thank you for your review and recommendation to accept our manuscript. We are grateful for your positive feedback and are pleased that our revisions have met your approval.
We appreciate the time and effort you have invested in reviewing our work and look forward to the prospect of our manuscript's publication in Sensor Networks.
Sincerely,
Jia
Reviewer 4 Report
Comments and Suggestions for Authors
Dear authors,
I have reviewed the new version of the manuscript and the responses to the 14 comments, and I consider that this improved version satisfies my comments and observations in the first stage of review.
Best Regards
Author Response

(The authors gave the same response as above.)

Reviewer 5 Report
Comments and Suggestions for Authors
The authors have mostly addressed the comments from the previous round of review and made sufficient revisions to the text. However, there are 2 requests.
1. The comment about the Gray code, 2421 code, and rem-3 code was not fully taken into account. Please provide the key references for these codes. In addition, the spelling of the code names in the text is inconsistent.
2. Is there an analogy in the Ponte::dispatch algorithm's operation with attention mechanisms of CNNs?
Author Response
Dear Reviewer,
Thank you very much for taking the time to review this manuscript. Please find the detailed
responses below and the corresponding revisions/corrections highlighted/in track changes in
the re-submitted files.
Q1: The comment about the Gray code, 2421 code, and rem-3 code was not fully taken into account. Please provide the key references for these codes. In addition, the spelling of the code names in the text is inconsistent.
R1: We are sorry about the confusion. We have searched for the details and references of the codes and added references to them. Besides, we have aligned the formal names with Wikipedia and changed the colloquial expression of the names of the binary codes (such as rem-3, 2421) in the previous manuscript to formal ones (Aiken Code, Stibitz). Additionally, we have added a short introduction to the code and experiment setup and highlighted in red in page 16; plus the inconsistent has been checked and corrected in the revision.
Q2: Is there an analogy in the Ponte::dispatch algorithm's operation with attention mechanisms of CNNs?
R2: Thank you for your insightful query regarding the potential analogy between the 'Ponte::dispatch' algorithm and attention mechanisms in CNNs. We did get inspiration from attention mechanisms, which are described in "SENet"[1]at the very beginning of our work from their intuition. 
While it does not directly implement an attention mechanism as traditionally defined in the context of CNNs, there are intuition similarities worth noting. As mentioned in the manuscript, because each of the input channels of the binary layer has only +1, or -1, it does not have the capability to "focus" on certain input channels, which is an inherent defect of BNNs; so what Ponte::dispatch trying to solve is exact to let the BNNs be able to focus on the importance of the different channel. SENet also aims to excite informative features in a class-agnostic manner to strengthen the shared low-level representations in early layers. 
However, there are differences between the two algorithms. From the role of the algorithm perspective, our Ponte::dispatch algorithm is designed to make the first binary layer aware of the associated weights from each pixel, but the goal of attention is to extract information from previous layers or do a class-specific manner. 
From the implementation view, the Ponte::dispatch using a mapping strategy which does not requires extra storage, and the attention continue to use the convolution like operation to solve the problem.
Therefore, while both concepts have some common in motivation, our approach lies in its ability to implementation, application areas, and problems to be solved, which is a different approach from the attention-based processing used in CNNs.
Best Regards,
Jia
[1] J. Hu, L. Shen and G. Sun, "Squeeze-and-Excitation Networks," 2018 IEEE/CVF Conference on Computer Vision and Pattern Recognition, Salt Lake City, UT, USA, 2018, pp. 7132-7141, doi: 10.1109/CVPR.2018.00745.